# Association of Japanese and Mediterranean Dietary Patterns with Muscle Weakness in Japanese Community-Dwelling Middle-Aged and Older Adults: *Post Hoc* Cross-Sectional Analysis

**DOI:** 10.3390/ijerph191912636

**Published:** 2022-10-03

**Authors:** Akio Shimizu, Kiwako Okada, Yasutake Tomata, Chiharu Uno, Fumiya Kawase, Ryo Momosaki

**Affiliations:** 1Department of Health Science, Faculty of Health and Human Development, The University of Nagano, 8-49-7, Miwa, Nagano 380-8525, Japan; 2Institute of Health and Nutrition, Nagoya University of Arts and Sciences, 57, Iwasaki-cho, Nisshin 470-0131, Japan; 3Graduate School of Nutritional Sciences, Nagoya University of Arts and Sciences, 57, Iwasaki-cho, Nisshin 470-0196, Japan; 4Department of Nutrition, Nagoya University of Arts and Sciences, 57, Iwasaki-cho, Nisshin 470-0196, Japan; 5School of Nutrition and Dietetics, Faculty of Health and Social Services, Kanagawa University of Human Services, 1-10-1, Heisei-cho, Yokosuka 238-8522, Japan; 6Department of Rehabilitation Medicine, Mie University Graduate School of Medicine, 2-174, Edobashi, Tsu 514-8507, Japan

**Keywords:** dietary patterns, nutritional epidemiology, geriatrics, sarcopenia, frailty

## Abstract

The association of Japanese and Mediterranean dietary patterns with muscle weakness in middle-aged and older Japanese individuals is unclear. This cross-sectional study investigated the association between Japanese and Mediterranean dietary patterns and muscle weakness in community-dwelling, middle-aged, and older Japanese individuals (enrolled from 2007 to 2011). Based on the dietary consumption information obtained from the brief self-administered diet history questionnaire, we assessed adherence to the Japanese (12-component revised Japanese diet index (rJDI12)) and Mediterranean (alternate Mediterranean diet (aMed) score) dietary patterns. Muscle weakness was defined as handgrip strength <28 and <18 kg for men and women, respectively. Logistic regression was used to ascertain the relationship between dietary pattern and muscle weakness. In our study, with 6031 participants, the Japanese, but not Mediterranean, dietary pattern was inversely associated with muscle weakness (*p* for trend = 0.031 and 0.242, respectively). In the model adjusted for confounders, including energy intake, the highest quartile of rJDI12 scores (9–12 points), and the rJDI12 scores, entered as continuous variables, showed an independent association (odds ratio (95% CI), 0.703 (0.507–0.974), and 0.933 (0.891–0.977), respectively). Our findings showed that adherence to the Japanese dietary pattern is associated with a low prevalence of muscle weakness.

## 1. Introduction

Epidemiological evidence shows that adherence to dietary patterns, such as Japanese and Mediterranean diets, decreases the incidence of coronary heart disease [1,2], disability [3,4], and all-cause mortality rate [5,6]. Various dietary patterns are associated with improved nutrient adequacy [7,8] and the prevention of poor outcomes. Therefore, adherence to these dietary patterns can be beneficial for health and longevity.

Frailty and sarcopenia constitute emerging health problems among the expanding older population in many countries and are increasingly associated with disability [9] and higher mortality [10]. The diagnostic criteria for frailty [11], dynapenia [12], and sarcopenia [13,14] include handgrip strength (HGS), which reflects the total muscle strength [15]. The Sarcopenia definition and outcomes consortium position statement emphasizes the importance of muscle weakness (low handgrip strength) as a component of sarcopenia, as it is associated with adverse health outcomes [14]. Muscle weakness is associated with insufficient dietary protein [16] and nutrient intakes [17,18]. Furthermore, adherence to the Mediterranean diet has been positively correlated with HGS in active, older Italian women [19]. Therefore, dietary patterns, as well as individual nutrient intake, may have an important role in maintaining muscle strength.

Stroke and dementia are common causes of functional disability in the Japanese population [20]. Adherence to the Japanese dietary pattern is associated with a decreased risk of dementia [21] and reduced incidence of functional disability [3]. Muscle weakness is known to cause functional disability through locomotion-related functional limitations [22]. Therefore, adherence to healthier dietary patterns and maintenance of muscle strength may effectively prolong disability-free survival with improved quality of life.

We hypothesized that individuals who adhere to healthier dietary patterns would have a lower prevalence of muscle weakness. Therefore, this study investigated the association between Japanese and Mediterranean dietary patterns and muscle weakness in community-dwelling middle-aged and older Japanese individuals.

## 2. Materials and Methods

### 2.1. Study Design

We conducted a secondary analysis of the data collected between 2007 and 2011 in a cross-sectional study of 7267 middle-aged and older adult participants of the Japan Survey on Aging (JSTAR) [16,23]. Briefly, the JSTAR is a panel study of nationally representative samples of Japanese residents aged ≥50 y in ten cities across Japan. Participants were randomly sampled from five municipalities in each target area. Four survey waves (2007, 2009, 2011, and 2013) were completed [16,23]. The JSTAR was conducted by the University of Tokyo, Hitotsubashi University and Research Institute of Economy, Trade, and Industry and obtained detailed data on the aspects of the daily lives of older adults, including their economic, social, and health conditions.

### 2.2. Dietary Survey

Dietary habits were evaluated using a previously validated food frequency questionnaire—the brief self-administered diet history questionnaire (BDHQ) [24]. The BDHQ includes 58 selected beverage and food items that are commonly consumed in Japan. This questionnaire records information based on the amount of food consumed in the month preceding the day of the survey. The daily intake of energy and specific nutrients was estimated using an ad hoc computer algorithm for the BDHQ [24,25]. Based on the results of the dietary survey, scores were calculated for Japanese and Mediterranean dietary patterns to determine their benefit in the maintenance of health status.

### 2.3. Japanese Dietary Pattern

We used the 12-component revised Japanese diet index (rJDI12) [26] to evaluate Japanese dietary patterns. The previously validated JDI12 comprises 12 items, except coffee, which was ranked as a nonbeneficial component [8,27]. However, owing to recent reports that high coffee intake has beneficial effects on health [28], the rJDI12 has reclassified coffee as a beneficial component. The rJDI12 score was calculated according to the intake of 12 food items that are included in the BDHQ: rice, miso soup, seafood (octopus, squid, shrimp, and clam; dried fish and salted fish; small fish with bones; canned tuna; oily fish; and non-oily fish), yellow and green vegetables (green leafy vegetables, including carrots, pumpkins, broccoli, tomatoes, tomato ketchup, boiled tomatoes, and stewed tomatoes), seaweed, pickles (salted green and yellow vegetable pickles and other salted vegetable pickles, excluding salted pickled plum), soybean products (bean curd, deep-fried bean curd), fruits (citrus, persimmon, and strawberry), mushrooms, green tea, pork, and beef (pork and beef, including ground pork and beef, ham, sausages). For 11 components (rice, miso soup, seafood, yellow and green vegetables, pickles, seaweed, soybean products, mushrooms, coffee, and green tea), one point was assigned if the intake was above that of the median sex difference; for one component (beef and pork), one point was assigned if the intake was below that of the median sex difference. Thus, the rJDI12 score ranged from 0 to 12, with high scores indicating increased conformity with the Japanese dietary pattern [26].

### 2.4. Mediterranean Dietary Pattern

The Mediterranean dietary pattern was evaluated using the alternate Mediterranean diet (aMed) score [29,30]. The aMed score is based on the Mediterranean diet score used in previous studies [31] and adapts traditional Mediterranean diet principles to non-Mediterranean countries. The aMed score comprises points for the following nine food items: whole grains, legumes, fish, fruits (including juices), vegetables (excluding potatoes), nuts, processed and red meats, monounsaturated fat-to-saturated fat ratio, and alcohol. However, the BDHQ does not include nut intake. Thus, the aMed score was calculated using eight items, excluding nuts, in this study. In addition, the BDHQ does not classify refined and unrefined grains; thus, the whole-grain score was calculated for all grains. The eight items were adjusted using the residual method, with total energy intake, based on previous studies [29]. Six components (whole grains, legumes, fish, vegetables (excluding potatoes), fruits (including juices), nuts, and monounsaturated fat-to-saturated fat ratio) were awarded one point for consumption that exceeded the median sex difference [31]. One component (red meat and processed meat) was awarded one point for consumption that was less than the median sex difference [31]. For the alcohol component, one point was awarded for consumption of 10–25 g/d for men and 5–15 g/d for women [30]. In this study, the aMed scores ranged from 0 to 8 points, with higher aMed scores indicating increased conformity with the Mediterranean dietary pattern [29].

### 2.5. Muscle Weakness

The HGS (kg) was used to assess muscle strength in this present study. Participants measured HGS according to the instructions of the interviewer. The HGS was measured in the dominant hand using a handgrip dynamometer (Smedley-type handgrip dynamometer, No. 6103, TANITA, Tokyo, Japan). The participants held the handgrip dynamometer as tightly as possible, with their arms parallel to the side of their body. Muscle weakness was assessed using the cutoff values (<28 kg for men and <18 kg for women), as proposed by the Asian Working Group for Sarcopenia 2019 [13].

### 2.6. Other Variables

Other variables included body mass index (BMI), history of comorbidities (cerebrovascular disease, coronary heart disease, diabetes, and cancer), history of alcohol consumption and smoking, time spent walking per day, and performance of instrumental activities of daily living (IADL). BMI was calculated by dividing the weight (kg) by the height (m) squared. The IADL were evaluated using the Tokyo Metropolitan Institute of Gerontology Index of Competence (TMIG-IC) [32]. The TMIG-IC consists of five items: shopping for daily necessities, using public transportation, paying bills, preparing meals, and handling bank accounts. The scores range from 0 to 5 points, with high TMIG-IC scores indicating high performance of IADL.

### 2.7. Statistical Analysis

Descriptive statistics were used to present data as numbers (percentages), mean (standard deviation), and median (interquartile range, IQR), wherever appropriate. A logistic regression model was used to calculate the odds ratios (OR), with 95% confidence intervals (CI), for muscle weakness, according to the quartiles of each dietary pattern score. The multivariate adjusted models were adjusted for different variables. Model 1 was adjusted for sex and age (50–54, 55–59, 60–64, 65–69, or ≥70 y). Model 2 was adjusted for BMI (<18.5, 18.5–25, ≥25 kg/m^2^, or missing) and the variables in Model 1 to determine whether the association between each dietary pattern and muscle weakness was affected by BMI. To determine whether the association between each dietary pattern and muscle weakness could be attributed to physical health status or other lifestyle factors, Model 3 was adjusted for IADL (TMIG-IC) score, smoking status (current, previous, never, or missing), alcohol consumption (every day, sometimes, never, or missing), time spent walking (≥1, 0.5–1, <0.5 h/d, or missing), and history of comorbidities (cerebrovascular disease, coronary heart disease, diabetes, cancer (yes, no, or missing for each term)). In addition, Model 4 was adjusted for only rJDI12 by adding energy intake (kcal/d; sex-specific quartile categories). Next, a sensitivity analysis was performed, and all models were entered into a logistic regression model, with each dietary pattern score as a continuous variable. Furthermore, we examined the association between each dietary pattern and muscle weakness, as stratified by sex. As a mediation analysis, Model 3 was adjusted for each nutrient of each dietary pattern, and their association with muscle weakness was analyzed. In addition, the association between combination of adherence to two dietary patterns (rJDI12 and aMed) and muscle weakness was analyzed. We classified each dietary pattern as a low score for Q1 and Q2 and high score for Q3 and Q4. Statistical analyses were performed using SPSS version 23.0 (IBM Japan, Tokyo, Japan). Statistical significance was set at *p* < 0.05.

### 2.8. Ethical Considerations

This study was approved by the Ethics Committee of Nagoya University of Arts and Sciences (approval number: 500). Written informed consent was obtained from each participant during recruitment for the JSTAR.

## 3. Results

In total, 7267 participants were enrolled; of these, 1296 had missing data. Hence, 6031 participants were included in the analysis (Figure 1). Of the 6031 participants, 53.6% were female, the mean age was 62.8 (7.0), and mean BMI (SD) was 23.1 (3.1) kg/m^2^. The quartiles of the two dietary patterns were slightly in agreement (Cohen’s kappa = 0.17, *p* < 0.001).

Table 1 compares the participant characteristics, according to the quartiles of each dietary pattern score. Participants with a high rJDI12 score were less likely to be smokers *(p* < 0.001). Additionally, participants with high rJDI12 scores tended to be older, walk ≥1 h/d, and have high energy intake (*p* < 0.001, respectively). Similarly, participants with high aMed scores were less likely to smoke (*p* = 0.001). In contrast, participants with high aMed scores were more likely to be male, have more than one form coronary heart disease, and be current drinkers (*p* < 0.001, respectively).

Appendix A show characteristics by gender in the quartiles. Only female participants showed differences in TMIG-IC scores by quartile of rJDI12 score (*p* = 0.036). Only men showed differences in alcohol consumption by quartile of aMed score (*p* < 0.001).

The nutrient intake through each dietary pattern is shown in Appendix A. High rJDI12 scores were associated with high nutrient intake. In contrast, high aMed scores tended to be associated with low intake of saturated fat and sugar.

The association between each dietary pattern and muscle weakness is shown in Figure 2, along with the ORs and 95% CIs. We found that a high rJDI12 score was inversely associated with the prevalence of muscle weakness (*p* = 0.031 for Model 4). This inverse association was not significantly different between men and women (*p* = 0.492 for interaction with sex). In contrast, a significant association was not observed between the aMed score and prevalence of muscle weakness. In the adjusted Model 4, the multivariate ORs (95% CIs) for the consecutive categories of rJDI12 scores were 1 (reference), 0.920 (0.711–1.190), 0.826 (0.630–1.1054), and 0.703 (0.507–0.974). Furthermore, when the rJDI12 score was entered as a continuous value, the OR was 0.933 (0.891–0.977).

The association between individual dietary pattern scores and muscle weakness, according to sex, is shown in Figure 3. In Model 4, the rJDI12 score showed no trend for either sex (*p* = 0.386 for men, *p* = 0.072 for women).

The results of mediation analysis are shown in Appendix A. The intermediate factors associated with the rJDI12 score and muscle weakness were protein (*p* = 0.058), dietary fiber (*p* = 0.265), vitamin C (*p* = 0.067), iron (*p* = 0.108), potassium (*p* = 0.292), and magnesium (*p* = 0.353).

The association between combination of adherence to two dietary patterns (rJDI12 and aMed) and muscle weakness is shown in Appendix A, along with the ORs and 95% CIs. Using both low scores as the reference (1.000), a high rJDI12, low aMed score 0.783 (0.625–0.982), and both high scores: 0.703 (0.507–0.974) were inversely associated with the prevalence of muscle weakness. However, there were no statistically significant associations between these groups and the prevalence of muscle weakness, when low rJDI12 and high aMed score were used as reference.

## 4. Discussion

This cross-sectional study investigated the association between the Japanese and Mediterranean dietary patterns and muscle weakness. The results showed that the rJDI12 score was inversely associated with the prevalence of muscle weakness. Notably, this study suggests that dietary patterns may influence muscle strength maintenance.

Adherence to Japanese dietary patterns may contribute to the maintenance of muscle strength, as seen in this study, where the participants with high rJDI12 scores had high energy intake. High energy intake is inversely related to the prevalence of muscle weakness in community-dwelling older adults [33]. Thus, energy intake may have a greater influence on muscle strength than dietary pattern. However, in the present study, muscle weakness was still inversely correlated with rJDI12 scores, even when energy intake was considered. Japanese dietary patterns are characterized by a high consumption of vegetables, fish, miso soup, seaweed, and green tea. Intake of vegetables, fish, and miso soup is associated with muscle strength [34,35,36]. Polyunsaturated fatty acids in fish are associated with beneficial effects on muscle lipids and mitochondrial function, which are major determinants of muscle function [37]. Therefore, the diet quality of the Japanese dietary pattern may positively influence muscle strength maintenance, regardless of energy intake.

Protein, dietary fiber, vitamin C, iron, potassium, and magnesium are intermediate factors of association between the Japanese dietary patterns and muscle weakness. These results are consistent with previous studies, showing association of muscle strength with nutrient intakes [37,38,39,40,41,42]. Additionally, aging [43] and inflammation [44] are related causes of muscle weakness. In our study, high rJDI12 scores were associated with older age, but inversely associated with muscle weakness. Dietary fiber [45], vitamin C [46], and magnesium [47] are nutrients that are associated with anti-inflammatory effects and may have a protective role against muscle weakness.

In contrast, the Mediterranean dietary pattern was not associated with muscle weakness in middle-aged or older Japanese individuals. This result differed from previous reports, which had shown that adherence to the Mediterranean dietary pattern was positively correlated with muscle strength in older Italian women [19]. These differences may be related to the dietary habits of the Japanese population. The consumption of vegetables, fruits, seafood, and legumes, which constitute the rJDI12, is reported to increase with age in Japan [1]. The rJDI12 may have sensitively reflected the differences in food intake in our study population. In addition, the consumption of whole grains and nuts, included in the aMed score, was low in the Japanese population [48] and may have affected the distribution of the aMed scores. Although there were limitations in applying the aMed score to these participants, there was only a slight agreement between the rJDI12 and aMed score. Therefore, the rJDI12 score, rather than the aMed score, may be a better parameter to reflect the dietary patterns of Japanese people. In the future, studies are needed to determine whether adherence to the Mediterranean dietary pattern is associated with the incidence of muscle weakness in the Japanese population on a longitudinal basis.

The strengths of this study were as follows: (1) this was a relatively large population-based cross-sectional study with 6031 participants; (2) the study included middle-aged and older people living in multiple regions of Japan; (3) the study compared Japanese and Mediterranean dietary patterns; and (4) the analysis was adjusted for many confounding factors.

However, this study had some limitations. First, as this was a cross-sectional study, the causal relationship between adherence to Japanese dietary patterns and muscle weakness remains unknown. However, we demonstrated that adherence to the Japanese dietary pattern resulted in a higher intake of some nutrients [37,38,39,40,41,42] that are associated with muscle strength. Second, this study did not include nuts in the calculation of aMed scores. Moreover, whole grains were not accurately graded. Furthermore, participants with high aMed scores had a high protein intake, but low saturated fat and sugar intakes, which may reflect the Mediterranean dietary pattern. Finally, not all the potential confounders were considered. For example, muscle strength is greatly influenced by exercise habits, such as resistance training [49]. Although we adjusted for daily walking time as a confounder in this study, the possibility of residual confounding factors cannot be ruled out. A longitudinal study may define the impact of specific dietary patterns on muscle strength in detail.

## 5. Conclusions

In conclusion, this cross-sectional study investigated the association between the Japanese and Mediterranean dietary patterns and muscle weakness. The results showed that adherence to the Japanese dietary pattern, but not the Mediterranean dietary pattern, was associated with a reduced prevalence of muscle weakness. Further research is crucial to determine whether adherence to the Japanese dietary pattern is associated with longitudinal muscle weakness.

Adherence to Japanese dietary patterns may contribute to the maintenance of muscle strength in the Japanese people.

## Figures and Tables

**Figure 1 ijerph-19-12636-f001:**
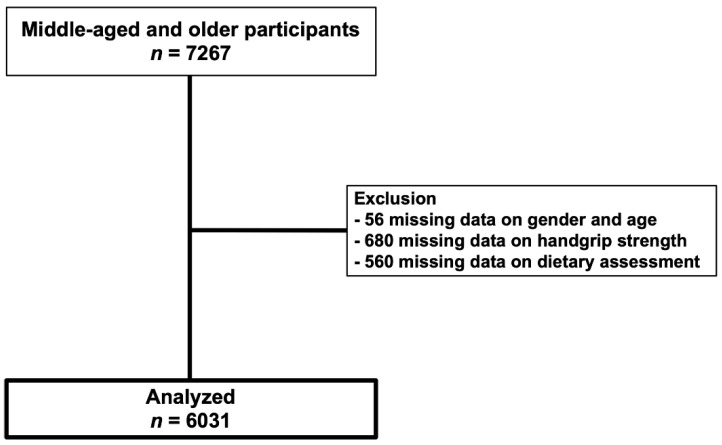
Flowchart depicting the selection of participants.

**Figure 2 ijerph-19-12636-f002:**
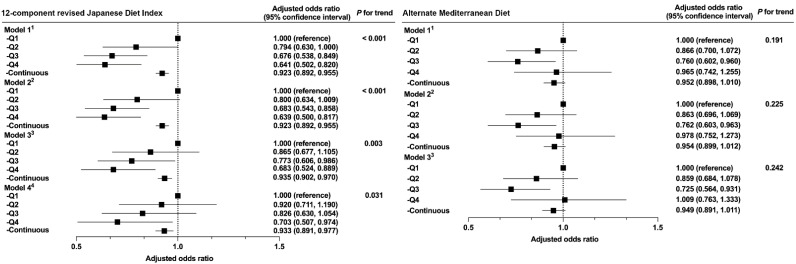
Relationship between the two dietary patterns and muscle weakness. ^1^ Model 1 was adjusted for age (50–54, 55–59, 60–64, 65–69, or ≥70 y) and sex. ^2^ Model 2 was adjusted for Model 1 plus BMI (<18.5, 18.5–25, ≥25 kg/m2, or missing). ^3^ Model 3 was adjusted for Model 2 plus IADL score (TMIG-IC score), time spent walking (≥1, 0.5–1, <0.5 h/d, or missing), alcohol consumption (every day, sometimes, never, or missing), smoking status (current, former, never, or missing), and history of diseases (cerebrovascular disease, coronary heart disease, diabetes, cancer (yes, no, or missing for each term)). ^4^ Model 4 was adjusted for Model 3 plus energy intake (kcal/d; sex-specific quintile categories). Continuous variables were entered into each model as continuous values for each dietary score.

**Figure 3 ijerph-19-12636-f003:**
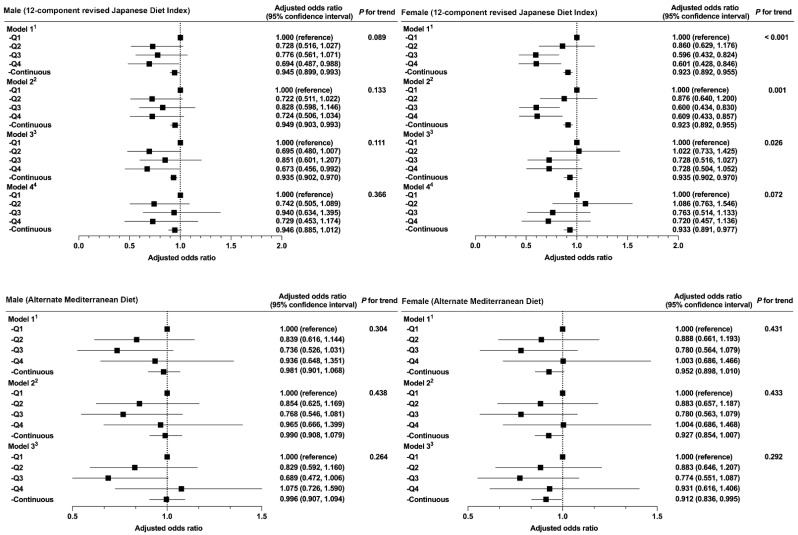
Relationship between the two dietary patterns and muscle weakness, stratified by sex. ^1^ Model 1 was adjusted for age (50–54, 55–59, 60–64, 65–69, or ≥70 y). ^2^ Model 2 was adjusted for Model 1 plus BMI (<18.5, 18.5–25, ≥25 kg/m2, or missing). ^3^ Model 3 was adjusted for Model 2 plus IADL score (TMIG-IC score), time spent walking (≥1, 0.5–1, <0.5 h/d, or missing), alcohol consumption (every day, sometimes, never, or missing), smoking status (current, former, never, or missing), and history of diseases (cerebrovascular disease, coronary heart disease, diabetes, cancer (yes, no, or missing for each term)). ^4^ Model 4 was adjusted for Model 3 plus energy intake (kcal/d; sex-specific quintile categories). Continuous variables were entered into each model as continuous values for each dietary score.

**Table 1 ijerph-19-12636-t001:** Characteristics of the participants across the two dietary patterns (*n* = 6031).

Characteristics	rJDI12 Group					aMed Group				
	Q1	Q2	Q3	Q4	*p*	Q1	Q2	Q3	Q4	*p*
Range of scores	0–4	5–6	7–8	9–12		0–3	4	5	6–8	
Participants, *n*	1581	1506	1674	1270		2706	1438	1203	684	
Age, years	61.3 (6.9)	62.3 (7.0)	63.4 (7.0)	64.5 (6.6)	<0.001	61.7 (7.0)	63.4 (7.0)	63.7 (6.8)	64.8 (6.6)	<0.001
Female sex, *n* (%)	835 (52.8)	801 (53.2)	892 (53.3)	705 (55.5)	0.488	1555 (57.5)	744 (51.7)	620 (51.5)	314 (45.9)	<0.001
BMI, kg/m2	23.2 (3.3)	23.1 (3.1)	23.2 (3.0)	23.0 (2.9)	0.288	23.0 (3.2)	23.2 (3.1)	23.2 (3.1)	23.3 (2.9)	0.157
Cerebrovascular diseases, *n* (%)					0.040					0.003
-Yes	49 (3.1)	40 (2.7)	40 (2.4)	42 (3.3)		65 (2.4)	42 (2.9)	41 (3.4)	23 (3.4)	
-No	962 (60.8)	929 (61.7)	784 (65.2)	458 (67.0)		1651 (61.0)	913 (63.5)	784 (65.2)	458 (67.0)	
-Missing	570 (36.1)	537 (35.7)	552 (33.0)	395 (31.1)		990 (36.6)	483 (33.6)	378 (31.4)	203 (29.7)	
Coronary heart diseases, *n* (%)					0.039					<0.001
-Yes	147 (9.3)	126 (8.4)	169 (10.1)	137 (10.8)		212 (7.8)	147 (10.2)	134 (11.1)	86 (12.6)	
-No	864 (54.6)	843 (56.0)	953 (56.9)	738 (58.1)		1504 (55.6)	808 (56.2)	691 (57.4)	395 (57.7)	
-Missing	570 (36.1)	537 (35.7)	552 (33.0)	395 (10.8)		990 (36.6)	483 (33.6)	378 (31.4)	203 (29.7)	
Diabetes, *n* (%)					0.033					<0.001
-Yes	143 (9.0)	162 (10.8)	160 (9.6)	125 (9.8)		221 (8.2)	163 (11.3)	127 (10.6)	79 (11.5)	
-No	869 (55.0)	807 (53.6)	962 (57.5)	750 (59.1)		1495 (55.2)	793 (55.1)	698 (58.0)	402 (58.8)	
-Missing	569 (36.0)	537 (35.7)	552 (33.0)	395 (31.1)		990 (36.6)	482 (33.5)	378 (31.4)	203 (29.7)	
Cancer, *n* (%)					0.095					0.001
-Yes	65 (4.1)	68 (4.5)	80 (4.8)	60 (4.7)		122 (4.5)	53 (3.7)	66 (5.5)	32 (4.7)	
-No	946 (59.8)	901 (59.8)	1042 (62.2)	815 64.2)		1594 (58.9)	902 (62.7)	759 (63.1)	449 (65.6)	
-Missing	570 (36.1)	537 (35.7)	552 (33.0)	395 (31.1)		990 (36.6)	483 (33.6)	378 (31.4)	203 (29.7)	
Smoking status, *n* (%)					<0.001					0.001
-Current	398 (25.2)	341 (22.6)	273 (16.3)	186 (14.6)		597 (22.1)	282 (19.6)	217 (18.0)	102 (14.9)	
-Former	376 (23.8)	343 (22.8)	439 (26.2)	316 (24.9)		620 (22.9)	360 (25.0)	299 (24.9)	195 (28.5)	
-Never	763 (48.3)	784 (52.1)	922 (55.1)	738 (58.1)		1422 (52.5)	756 (52.6)	662 (55.0)	367 (53.7)	
-Missing	44 (2.8)	38 (2.5)	40 (2.4)	30 (2.4)		67 (2.5)	40 (2.8)	25 (2.1)	20 (2.9)	
Alcohol consumption, *n* (%)					0.108					<0.001
-Every day	346 (21.9)	334 (22.2)	384 (22.9)	232 (18.3)		558 (20.6)	331 (23.0)	238 (19.8)	169 (24.7)	
-Sometimes	538 (34.0)	513 (34.1)	564 (33.7)	476 (37.5)		891 (32.9)	489 (34.0)	449 (37.3)	262 (38.3)	
-Never	645 (40.8)	609 (40.4)	666 (39.8)	508 (40.0)		1166 (43.1)	569 (39.6)	469 (39.0)	224 (32.7)	
-Missing	52 (3.3)	50 (3.3)	60 (3.6)	54 (4.3)		91 (3.4)	49 (3.4)	47 (3.9)	29 (4.2)	
Time spent walking, *n* (%)					<0.001					0.120
-≥1 h/d	571 (36.1)	618 (41.0)	753 (45.0)	593 (46.7)		1079 (39.9)	620 (43.1)	521 (43.3)	315 (46.1)	
-0.5–1 h/d	568 (35.9)	558 (37.1)	601 (35.9)	443 (34.9)		1006 (37.2)	505 (35.1)	439 (36.5)	220 (32.2)	
-<0.5 h/d	397 (25.1)	306 (20.3)	290 (17.3)	212 (16.7)		564 (20.8)	287 (20.0)	219 (18.2)	135 (19.7)	
-Missing	45 (2.8)	24 (1.6)	30 (1.8)	22 (1.7)		57 (2.1)	26 (1.8)	24 (2.0)	14 (2.0)	
TMIG-IC score, points	5 (5–5)	5 (5–5)	5 (5–5)	5 (5–5)	0.036	5 (5–5)	5 (5–5)	5 (5–5)	5 (5–5)	0.099
Energy, kcal/d	1441 (449)	1778 (474)	2035 (536)	2409 (633)	<0.001	1847 (629)	1901 (644)	1939 (600)	1986 (605)	<0.001

Data are expressed as numbers (percentage), mean (SD), or median (IQR: 25%–75%), unless otherwise indicated. aMed, alternate Mediterranean diet; BMI, body mass index; rJDI12, 12-component revised Japanese diet index; TMIG, Tokyo Metropolitan Institute of Gerontology Index of Competence.

## Data Availability

Data cannot be shared for privacy or ethical reasons.

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
