# Peer review of "Association of Japanese and Mediterranean Dietary Patterns with Muscle Weakness in Japanese Community-Dwelling Middle-Aged and Older Adults: Post Hoc Cross-Sectional Analysis"

_ijerph, 2022, doi:10.3390/ijerph191912636_

Round 1
Reviewer 1 Report
A) In line 193, the author states the Females were 44.5%, but in Table 1, the average female number is above this value. Can authors correct this mistake?
B) Authors should segregate disease rates, smoking habits, etc into three categories: Male/Female/Total data for each diet group in Table 1. Diets may affect genders differently and should be identified in this analysis.
C) A Venn diagram showing overlap between the diet group for rJDI12 and a Med group will be beneficial for the readers to understand the data..
Author Response
September 26, 2022
Prof. Dr. Paul B. Tchounwou
Editor-in-Chief
International Journal of Environmental Research and Public Health
Dear Editor
I, along with my coauthors, would like to re-submit the revised manuscript entitled “Association of Japanese and Mediterranean dietary patterns with muscle weakness in Japanese community-dwelling middle-aged and older adults: Post hoc cross-sectional analysis.” for publication in International Journal of Environmental Research and Public Health as an Original Article. The Manuscript ID is ijerph-1928551.
The manuscript has been carefully rechecked, and appropriate changes have been made in accordance with the reviewer’s comments. Changes to the text according to these comments have been marked in red font. Our responses to the reviewer’s comments have been prepared and attached herewith.
We thank you and the reviewers for your thoughtful suggestions and insights, which have enriched our manuscript, resulting in a more balanced and better account of our research. We hope that the revised manuscript is now suitable for publication in your journal.
I look forward to your reply.
Sincerely,
Yasutake Tomata, RD, PhD
School of Nutrition and Dietetics, Faculty of Health and Social Services, Kanagawa University of Human Services, 1-10-1, Heisei-cho, Yokosuka, Kanagawa, 238-8522, Japan
Tel: +81-46-828-2500
E-mail: [email protected]
Authors’ Responses to the Reviewers’ Comments
Reviewer 1.
We thank the reviewer for the helpful comments, which assisted us in improving the manuscript. We have made additional changes in the revised manuscript based on the comments. Our changes have been marked in red font in the revised manuscript.
Comment #1: In line 193, the author states the Females were 44.5%, but in Table 1, the average female number is above this value. Can authors correct this mistake?
Response: We agree with the reviewer’s comment. We apologize for the confusion caused by the inaccurate value. Therefore, we have revised the relevant sentence in the Results of the manuscript as follows:
“Of the 6,031 participants, 53.6% were female, the mean age was 62.8 (7.0), and the mean BMI (SD) was 23.1 (3.1) kg/m2.” (Page 4, line 176)
Comment #2: Authors should segregate disease rates, smoking habits, etc into three categories: Male/Female/Total data for each diet group in Table 1. Diets may affect genders differently and should be identified in this analysis.
Response: We agree with the reviewer’s comment. Accordingly, we have added the relevant sentence to the Results section as follows:
“Supplemental Table 1 (rJDI12 score) and 2 (aMed score) show characteristics by sex in the quartiles. Only female participants showed differences in TMIG-IC scores by quartile of rJDI12 score (P=0.036). Only men showed differences in alcohol consumption by quartile of aMed score (P<0.001).” (Page 7, lines 196-199)
Comment #3: A Venn diagram showing overlap between the diet group for rJDI12 and a Med group will be beneficial for the readers to understand the data.
Response: We thank the reviewer for the comment. In this study, dietary patterns were classified by quartiles. Thus, a Venn diagram for the eight patterns would be complex. Therefore, we used Cohen's Kappa to identify the agreement of the quartiles in the two dietary patterns. We added the relevant text to the Results section as follows:
“The quartiles of the two dietary patterns were slightly in agreement (Cohen's kappa=0.17, P<0.001).” (Page 4, lines 177-179)

Reviewer 2 Report
Dear authors and Editor,
First, I am glad for the opportunity to review this interesting manuscript titled “Association of Japanese and Mediterranean dietary patterns with muscle weakness in Japanese community-dwelling middle-aged and older adults: Post hoc cross-sectional analysis”. This is a cross-sectional study that analyze the dietary effects on muscle weakness in participants > 50 years old.
##General comments to authors: The research topic is very interesting and represents a first step to establish relationships between diet habits and muscle weakness. In general, all sections of the manuscript are suitable, so I congrats the authors. The only point that in my viewpoint needs to be modify is that the study is more focus on Japanese than in Mediterranean. I think a more balance information could help readers to understand differences between this two. I hope next comment help authors to improve this manuscript.
Abstract:
# Comment 1: lines 28-30 seems a little confusing to me, rewrite them to help readers.
Introduction:
# Comment 1: lines 48-51 Is this reference correct?
# Comment 2: line 57. I do not see the point of this statement, please provide a clearer relation to the research topic. Thanks.
Material and Methods:
# Comment 1: I congrats the authors for managing this huge sample size, this provide sound results.
# Comment 2: Although this sample was extracted from previous study, please explained how the recruitment process was performed.
# Comment 3: section 2.3 and 2.4 are very clarifying, thanks.
# Comment 4: Who carried out the assessment of HGS?
Results:
# Comment 1: Were there significant differences between groups at baseline? Baseline characteristics has been narratively described, but I still missing if they differ statistically. Because if it is true, this is a limitation of study that should be mentioned in limitation section.
# Comment 2: Could authors please write a paragraph in which comparison results between Japanese or Mediterranean dietary are clearer? Please.
# Comment 3: Sensitive analysis is very sound, congrats.
Discussion:
# Comment 1: Please, provide higher information of why Mediterranean dietary did not seem to be related to HPG. This issue needs to be further study?
Conclusion:
# Comment 1: Please, provide more information about Mediterranean dietary.
# Comment 2: The acknowledgment section should be change or remove, please.
Supplementary material:
#Comments Table S1: It is very illustrative I congrats authors, but where are comparison values? Maybe authors could (1) add more p-values or (2) add mean differences values. I think if authors choose second option, they could create another table.
Author Response
September 26, 2022
Prof. Dr. Paul B. Tchounwou
Editor-in-Chief
International Journal of Environmental Research and Public Health
Dear Editor
I, along with my coauthors, would like to re-submit the revised manuscript entitled “Association of Japanese and Mediterranean dietary patterns with muscle weakness in Japanese community-dwelling middle-aged and older adults: Post hoc cross-sectional analysis.” for publication in International Journal of Environmental Research and Public Health as an Original Article. The Manuscript ID is ijerph-1928551.
The manuscript has been carefully rechecked, and appropriate changes have been made in accordance with the reviewer’s comments. Changes to the text according to these comments have been marked in red font. Our responses to the reviewer’s comments have been prepared and attached herewith.
We thank you and the reviewers for your thoughtful suggestions and insights, which have enriched our manuscript, resulting in a more balanced and better account of our research. We hope that the revised manuscript is now suitable for publication in your journal.
I look forward to your reply.
Sincerely,
Yasutake Tomata, RD, PhD
School of Nutrition and Dietetics, Faculty of Health and Social Services, Kanagawa University of Human Services, 1-10-1, Heisei-cho, Yokosuka, Kanagawa, 238-8522, Japan
Tel: +81-46-828-2500
E-mail: [email protected]
Authors’ Responses to the Reviewers’ Comments
Reviewer 2.
We thank the reviewer for the helpful comments in the previous review, which helped improve the manuscript. We have made additional changes based on the comments. Our changes have been marked in red font in the revised manuscript.
Abstract:
Comment #1: lines 28-30 seems a little confusing to me, rewrite them to help readers.
Response: We thank the reviewer for this comment on the ambiguity in lines 28-30. We have revised the relevant sentence in the Abstract as follows:
Introduction:
Comment #2: lines 48-51 Is this reference correct?
Response: We have corrected the Reference.
Comment #3: line 57 I do not see the point of this statement, please provide a clearer relation to the research topic. Thanks.
Response: We agree with the reviewer’s comment. We thought it would be easier to understand the purpose of the study if we removed the sentence pointed out by the reviewers.
Therefore, the 'However, to the best of our knowledge, the association between adherence to healthier dietary patterns and muscle weakness has not been investigated yet.' has been deleted.
Material and Methods:
Comment #4: Although this sample was extracted from previous study, please explained how the recruitment process was performed.
Response: We agree with the reviewer’s comment on the need to explain how participants were sampled. Therefore, we have added the relevant sentence to the Methods section as follows:
“Participants were randomly sampled from five municipalities in each target area.” (Page 2, lines 66-67)
Comment #5: Who carried out the assessment of HGS?
Response: We agree with the reviewer’s comment. As the reviewer pointed out, it is essential to provide information on how the HGS measurement was performed by the participant. In this study, participants performed HGS measurements as instructed by the interviewer. The interviewer recorded the results.Therefore, we have added the relevant sentence to the Methods section as follows:
“Participants measured HGS according to the instructions of the interviewer.” (Page 3, lines 125-126)
Results:
Comment #6: Were there significant differences between groups at baseline? Baseline characteristics has been narratively described, but I still missing if they differ statistically. Because if it is true, this is a limitation of study that should be mentioned in limitation section.
Response: We agree with the reviewer’s comment. As pointed out, showing whether there were significant differences in each result is important. Therefore, we have added the p values in the text and tables.
Comment #7: Could authors please write a paragraph in which comparison results between Japanese or Mediterranean dietary are clearer? Please.
Response: We agree with the reviewer’s comment. We classified rJDI12 and aMed scores into low or high scores in the quartiles, and the association between each score combination and the prevalence of muscle weakness was examined in a logistic regression model. However, when low rJDI12 and high aMed scores were used as reference, all combinations of dietary scores showed no significant differences. Therefore, there was no clear difference between the two dietary patterns;however, adherence to the Japanese dietary pattern may reduce the prevalence of muscle weakness. Therefore, we have added a relevant sentence to the Methods and Results section as follows:
“In addition, the association between combination of adherence to two dietary patterns (rJDI12 and aMed) and muscle weakness was analyzed. We classified each dietary pattern as a low score for Q1 and Q2 and a high score for Q3 and Q4.” (Page 4, lines 165-167)
and
“The association between combination of adherence to two dietary patterns (rJDI12 and aMed) and muscle weakness is shown in Supplemental table 6, along with ORs and 95% CIs. Using both low scores as the reference (1.000), a high rJDI12 and low aMed score: 0.783 (0.625-0.982), and both high scores: 0.703 (0.507-0.974) were inversely associated with the prevalence of muscle weakness. However, there were no statistically significant associations between these groups and the prevalence of muscle weakness when low rJDI12 and high aMed score were used as reference.” (Page 8, lines 242-247)
In addition, we have added Supplemental table 6.
Discussion:
Comment #8: Please, provide higher information of why Mediterranean dietary did not seem to be related to HGS. This issue needs to be further study?
Response: We agree with the reviewer’s comment. We believe that further explanation of the non-association between Mediterranean diet pattern (aMed score) and muscle weakness in the study population is needed. In this study population, the rJDI12 score and aMed score were in slight agreement (Cohen's kappa=0.17, P<0.001). Therefore, we consider the rJDI12 to be a more reflective indicator of the dietary habits of the Japanese population. In addition, as the reviewer points out, further longitudinal studies are needed on the aMed score and incidence of muscle weakness. Therefore, we have added a relevant sentence to the Discussion section as follows:
“The rJDI12 may have sensitively reflected the differences in food intake in our study population. In addition, the consumption of whole grains and nuts included in the aMed score was low in the Japanese population [48], and may have affected the distribution of the aMed scores. Although there were limitations in applying the aMed score to these participants, there was only a slight agreement between the rJDI12 and aMed score. Therefore, the rJDI12 score, rather than the aMed score, may be a better parameter to reflect the dietary patterns of Japanese people. In the future, studies are needed to determine whether adherence to the Mediterranean dietary pattern is associated with the incidence of muscle weakness in the Japanese population on a longitudinal basis.” (Page 9, lines 279-287)
Conclusion:
Comment #9: Please, provide more information about Mediterranean dietary.
Response: As the reviewer noted, it is important to show in the Conclusions section that the Mediterranean dietary pattern was not associated with a lower prevalence of muscle weakness. Therefore, we have added a relevant sentence to the Conclusions section as follows:
“The results showed that adherence to the Japanese dietary pattern, but not the Mediterranean dietary pattern, was associated with a reduced prevalence of muscle weakness.” (Page 10, lines 307-309)
Comment #10: The acknowledgment section should be change or remove, please.
Response: We thank the reviewer for the suggestion. We have removed the Acknowledgements section.
Supplementary material:
Comment #11: It is very illustrative I congrats authors, but where are comparison values? Maybe authors could (1) add more p-values or (2) add mean differences values. I think if authors choose second option, they could create another table.
Response: We agree with the reviewer’s comment. We agree that it is important to show the significant differences in the supplemental tables. Therefore, we have added p values to our supplemental table.

Round 2
Reviewer 1 Report
The current draft of manuscript can be accepted after minor grammar and spell check.